



# Modeling the annual cycle of daily Antarctic sea ice extent

Mark S. Handcock[1] and Marilyn N. Raphael[2]

[1]Department of Statistics, UCLA
[2]Department of Geography, UCLA

**Correspondence:** Mark S. Handcock (handcock@stat.ucla.edu)

**Abstract.** The total Antarctic sea ice extent (SIE) experiences a distinct annual cycle, peaking in September and troughing in March. In this paper we propose a mathematical and statistical decomposition of this temporal variation in SIE. Each component is interpretable and, when combined, give a complete picture of the variation of the sea ice. We consider time scales varying from the instantaneous, and not previously defined, to the multidecadal curvilinear trend, the longest. Because our representation is daily, these timescales of variability give precise information about the timing and rates of advance and retreat of the ice and may be used to diagnose physical contributors to variability in the sea ice. We define a number of annual cycles each capturing different components of variation, especially the yearly amplitude and phase that are major contributors to SIE variation. Using daily sea ice concentration data, we show that our proposed invariant annual cycle explains 29% more of the variation of daily SIE than the traditional method. The proposed annual cycle that incorporates amplitude and phase variation explains 77% more variation than the traditional method. The variation in phase explains more of the variability in SIE than the amplitude. Using our methodology, we show that the anomalous decay of sea ice in 2016 was associated largely with a change of phase rather than amplitude. We show that the long term trend in Antarctic sea ice extent is strongly curvilinear and the reported positive linear trend is small and dependent strongly on a positive trend that began around 2011 and continued until 2016.

## 1 Introduction

Much of the research on Antarctic sea ice variability focuses on the monthly, seasonal and interannual time scales (Parkinson and Cavalieri, 2012; Turner et al., 2015b; Simpkins et al., 2012; Hobbs et al., 2015; Holland, 2014; Holland et al., 2017). This is useful and necessary, especially if links to the larger scale (and remote) atmospheric and oceanic forcings are to be made. However, significant aspects of the timing of the ice cycle, for example when ice advance or ice retreat begins, occur at sub-monthly scales (Stammerjohn et al., 2008; Turner et al., 2017; Stuecker et al., 2017; Schlosser et al., 2018; Meehl et al., 2019). Using daily data facilitates analysis of the daily variation of sea ice and is the springboard of this research.

The dominant/primary characteristic of Antarctic sea ice variability is its annual cycle. Satellite-observed, total Antarctic sea ice extent (SIE) experiences a distinct annual cycle, peaking in September (19 million km$^2$) and troughing in late February (3 million km$^2$) on average. The growth from minimum to maximum (peak) is slower than the retreat from maximum to minimum (trough). This is arguably the strongest seasonal cycle on the planet. The amplitude and phase of the annual cycle also vary regionally (Raphael and Hobbs, 2014). The daily, annual cycle of SIE is traditionally calculated by simply taking the average





(or the median value) for each day of the year. However, satellite-observed SIE can vary widely from day to day. Some of this variation is due to the ice growth, melting and divergence of the ice at the ice edge, while some is due, for example, to transient effects of cloud, and melt on the ice surface (e.g., Comiso and Steffen, 2001). A simple daily average or median includes

all of these sources of variability, perhaps leading to over/under estimation of the SIE. Therefore, a standard deviation (or a percentile) is often included to give some idea of the variability of the individual days around the mean for that day. While simple and transparent, this method of calculating the annual cycle produces a value that is subject to substantial variation since it is based on as few as 40 numbers (the length of the satellite observed data time-series), one for each year of recorded data, and does not include the effect of the day preceding nor the day following the averaged day. It is also influenced by the pattern

of missing values. Finally, it also disguises the fact that the daily annual cycle might be slowly changing phase and that the amplitude as well as shape of the daily annual cycle of SIE might vary. This can make it difficult to make statistically sound conclusions about variability in the data.

The need for accurate representation of the annual cycle is not limited to SIE data. There have been a number of studies that have examined the annual/seasonal cycle of other climate variables (Stine et al., 2009, e.g.,). The limitations of the traditional

method of calculating the annual cycle have also been recognized, for example by Deng and Fu (2018) who evaluated several methods for extracting the annual cycle from climate data. Our overarching aim in this research is not only to redefine the annual cycle, but also to make a meaningful decomposition of the variation of the annual cycle of Antarctic SIE. We do so on the time dimension in such a way that each component can be interpreted individually and, when taken together, all of the components give a complete picture of the variation of the sea ice. We consider the variation from the shortest timescale,

instantaneous variation, increasing the timescale sequentially we move through the day-to-day variation, the year-to-year (interannual) variation, and finally the longest timescale, the curvilinear trends of the multi-decadal variation. In the process, we make a number of technical contributions, most importantly to define complementary types of annual cycles that are meaningful in terms of this decomposition, and also to the representation of volatility. We have deliberately chosen (time) dimensions based on their interpretability rather than solely statistical efficiency concerns. For example, the amplitude and phase components of

the decomposition are much more interpretable than simple spectral components.

We begin by presenting a model that allows the mathematical and stochastic representation of the proximate forces that lead to the recorded annual cycle of sea ice extent. These mathematical and stochastic methods incorporate real variability in SIE and reduces the contribution from the ephemeral effects described above. They also allow amplitude and phase dilation and contraction. Thus, the annual cycle is not constrained to be a fixed cyclical pattern rather, it is a pattern that allows both temporal

dilation and contraction as well as amplitude modulation. To show the utility of the model, we develop several different annual cycles including one that is invariant, one that is adjusted for phase only and one that is adjusted for amplitude only. From the modeled annual cycles we define and extract the variability at the timescales mentioned. We conclude with a decomposition of the variability of SIE during 2016, the year of anomalous decay of SIE. The data are described in Sect. 2, the model is defined and developed in Sect. 3. The results are presented and discussed in Sect. 4 while concluding remarks are presented in Sect. 5.

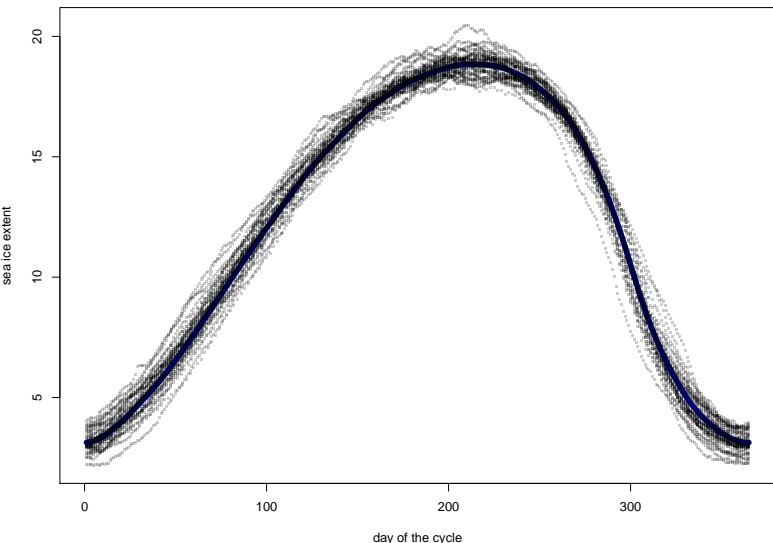

**Figure 1.** Recorded Sea Ice Extent (SIE) (grey) for each year, compared to a smooth Annual Cycle (blue) over a 365 day period. The horizontal axis is the day of the cycle and the vertical axis is sea ice extent in millions of km$^2$.

## 2   Data

This study uses sea ice concentration (SIC) data from the Bootstrap Sea Ice Concentrations from Nimbus-7 SMMR and DMSP SSM/I-SSMIS, Version 3 (Comiso, 2017; Peng et al., 2013; Meier et al., 2017). These data were generated using the Advanced Microwave Scanning Radiometer - Earth Observing System (AMSR-E) Bootstrap Algorithm with daily varying tie-points. They span the period 26 October 1978 to 31 December 2018 and are daily except prior to July 1987 when they are every other day. Data are gridded on the SSM/I polar stereographic grid (25 x 25 km). In addition to the alternate day observations from 1978-1987, there are a number of days and segments of days with no observations. Our methods do not require a complete temporal data record and naturally deal with missing data. The SIE used in our analysis was calculated using the conventional limit of the 15% SIC isoline. Every grid poleward of the 15% isoline is considered to be completely covered with ice.

Fig. 1 shows the recorded total SIE (in grey dots) for each year from 1979 - 2017 and a smoothed representation of the traditional daily annual cycle (blue). In this figure, day 0 on the horizontal axis represents the lowest SIE for the year, typically occurring around Julian day 50. We employ this convention for all of the time-series figures used in this paper. The plot illustrates nicely the variation of the SIE from day-to-day and also from year-to-year.

## 3   Methods and results: A statistical decomposition of sea ice extent

### 3.1   Annual cycle definition

*Traditional Annual Cycle*



Our decomposition of the sea ice extent starts with the traditional representation based on the annual cycle is:

$$\text{extent}(t) = a[\text{doy}(t)] + \alpha(t) \qquad \text{where} \quad t = T_0, \ldots, T \tag{1}$$

where extent(t) is the extent on day t expressed as a decimal year (e.g, Feb 1, 2010 as 2010.08767), $\text{doy}(t)$ is the day of the year for t (e.g., 32). Most importantly, $a$ is an *annual cycle shape function* with $a(s)$ giving the annual cycle shape value for
day-of-the-year s. In this context, $\alpha(t)$ is the *anomaly* of the extent from the annual cycle on day $t$, $T_0$ is the first observed time and $T$ is the last observed time. For the data in this paper, $T_0 = 1978.833$ and $T = 2019$.

Within this representation, the annual cycle is traditionally estimated by $a_T[s]$:

$$a_T[s] = \frac{1}{\sum_{t:\text{doy}(t)=s} 1} \sum_{t:\text{doy}(t)=s} \text{extent}(t) \qquad \text{where} \quad t = T_0, \ldots, T \tag{2}$$

where $\sum_{t:\text{doy}(t)=s} 1 = 40$ is the number of years of data.
This traditional estimate, $a_T[s],$) has a number of statistical issues which reduce its utility for examining the sea ice variability. Firstly, it is typically based on data for a subset of the satellite era (e.g, from 1979 forward). Currently, this is about forty years of data, inducing intrinsic statistical variability into $a_T[s]$ as an estimate of $a[s]$. This could be reduced by increasing the temporal range backward, by, for example, including data from the earlier satellite record (NIMBUS-5). Another option is to include information from proxy sources. We do not further consider these in this paper. Secondly, $a_T[s]$ is computed
separately for each day, ignoring the surrounding days. There is information in the temporally close days in the intuitive sense that days close to s, e.g., $s-1$ and $s+1$ will have similar values, albeit not exactly the same. This information is ignored by $a_T[s]$. Thirdly, we expect $a[s]$) to be smooth as a function of $s$ so that changes in $a_T[s]$ with $s$ will be similar for days that are close. Fourthly, we expect that $a_T[s]$ will "*over fit*" to the record making the estimated anomalies from it smaller than the true anomaly, $\alpha(t)$, and the annual cycle estimates will be more variable than the true annual cycle. This last issue is induced
by the finite record and the estimates of the anomaly $\hat{\alpha}(t) = \text{extent}(t) - a_T[\text{doy}(t)]$ will be statistically different than those of $\alpha(t)$. In sum, the traditional estimate, $a_T[s]$, uses limited information, ignores other days, is not as smooth as we expect, due to day-to-day variation and it over fits to the record.

*Invariant Annual Cycle*

It is possible that smoothing the data could be a solution to the statistical issues that arise from the way in which the traditional
annual cycle is calculated. To address this we define an *invariant annual cycle*, $a_I[s]$, which models $a[s]$ as a cyclic cubic spline function (Wegman and Wright, 1983) of $s$. Specifically, $a[s]$ is modeled as a piece-wise cubic polynomial that has a continuous second derivative, is continuous, has continuous 1st and 2nd derivatives at $T$ and best fits the recorded (satellite-observed) extents while being smooth. The specific criterion for the last feature is to choose $a_I[s]$ to minimize the penalized square error (PSE):

$$\text{PSE}_\lambda(a) = \sum_{t=T_0}^{T} \{\text{extent}(t) - a[\text{doy}(t)]\}^2 + \lambda \int_0^{365} a''[s]^2 ds \qquad \lambda > 0 \tag{3}$$





where $a''[s]$ is the 2$^{\text{nd}}$ derivative of $a[s]$ and $\lambda$ is a smoothing parameter, chosen to balance the closeness of fit to the recorded values (the first term) with the smoothness of $a[s]$ (the second term). Hence, choosing the function $a[s]$ that minimizes $\text{PSE}_\lambda(a)$ provides a balanced representation of the annual cycle. It prioritizes smoothness of $a[s]$ over the closeness of fit of $a[s]$ to the recorded extents. Note that the traditional estimator, $a_T[s]$, is the minimizer with $\lambda = 0$, that is, with no penalty for lack of

smoothness. The choice of $\lambda$ is subjective. In this work we choose to maximize the ability to predict unrecorded extents. Specifically, we use Generalized Cross Validation (GCV) (Craven and Wahba, 1978) to choose, and the R package `mgcv` by Simon Wood for analysis (Wood, 2004, 2017). The annual cycle so obtained is the optimal smoothest annual cycle chosen to minimize the mean squared error (MSE) of SIE. Any trends are removed and there is no adjustment for phase or amplitude. Fig. 2(a) compares the traditional annual cycle (plotted from Julian day 50 in 2016 to day Julian day 49 in 2017), with the

recorded SIE, and the invariant annual cycle. The visual improvement is modest but, as shown in Table 1, the invariant annual cycle represents a 29.8% improvement in the MSE compared to the traditional. Note that both annual cycles overestimate the SIE in the retreat phase of the ice.

*Amplitude and Phase Adjusted Annual Cycle*

The invariant annual cycle has the same motivation as the traditional annual cycle while being a clear statistical and conceptual

improvement over the traditional. However, we argue that since it is also fixed by day-of-year, it may be too restrictive since it, like the traditional, disguises the contributions of both amplitude and phase to the annual cycle. To address this we define a complementary annual cycle that is deformed each year in two ways. The first is *amplitude* in the sense that the yearly maximum and minimum extents may vary but the *shape* of the daily extent may be invariant. We enable the annual cycle to vary from year-to-year as a parametrized function of the annual cycle shape function. Specifically, we define the *amplitude*

*adjusted annual cycle*, $a_A[s, y]$ to satisfy:

$$\text{extent}(t) = a_A[\text{doy}(t), \text{min}\cdot\text{extent}(\text{year}(t)), \text{max}\cdot\text{extent}(\text{year}(t))] + \alpha(t) \tag{4}$$

where

$$a_A[s, \text{min}, \text{max}] = u_A[s](\text{max} - \text{min}) + \text{min} \tag{5}$$

and year(t) is the year for t (e.g., 2010), max·extent$(y)$ is the scale parameter giving the maximum extent for year y and

min·extent$(y)$) is the scale parameter giving the minimum extent for year y. Here $u_A[s]$ is an invariant annual cycle for the standardized extent. It is defined in an analogous way to the invariant annual cycle as a smooth function. Specifically, $u_A[s]$ as a cyclic cubic spline function of $s$ chosen to minimize the penalized square error:

$$\text{PSE}_{\lambda_A}(u) = \sum_{t=T_0}^{T} \left\{ \frac{\text{extent}(t) - \text{min}\cdot\text{extent}(\text{year}(t))}{\text{max}\cdot\text{extent}(\text{year}(t)) - \text{min}\cdot\text{extent}(\text{year}(t))} - u[s] \right\}^2 + \lambda_A \int_0^{365} u''[s]^2 ds \qquad \lambda_A > 0 \tag{6}$$

where $\lambda_A$ is a smoothing parameter with the same role as $\lambda_I$.

This annual cycle gives a different decomposition of the extent than the invariant annual cycle as it captures variation due to amplitude variation. Specifically, adjusting for amplitude results in a 55.2% improvement in the MSE compared to the





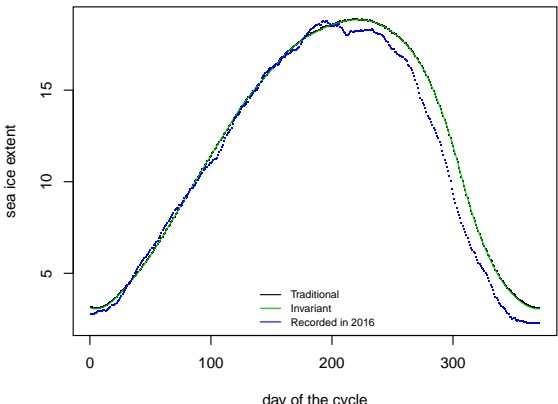
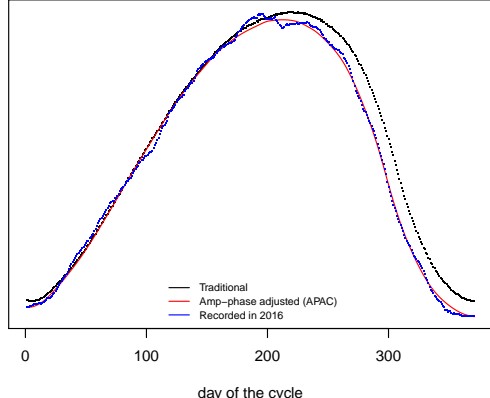

**Figure 2.** Comparison of Annual Cycle estimates: Panel (a) Traditional and Invariant; Panel (b) traditional and Amp-phase adjusted. The horizontal axis is the day of the cycle and the vertical axis is sea ice extent in millions of km$^2$.

traditional (See Table 1). Note that this allocates that component of the variation in extent due to amplitude variation to the annual cycle rather than the residual term, $\alpha(t)$ (See Eq. (4)). The magnitude of the change clearly underscores the importance of amplitude variations in the definition of the annual cycle.

Another component of the annual cycle that is important is the phase. This is the timing of the maximum and minimum extents. It is important because it determines the length of the annual cycle and influences its shape. We enable the annual cycle to vary from year-to-year as a parametrized function of the phase of the annual cycle shape function, defining the *phase adjusted annual cycle*, $a_P[s]$:

$$\text{extent}(t) = a_P[\text{phase}(t)] + \alpha(t) \tag{7}$$

where phase$(t)$ is the phase-adjusted day of the year for t (e.g., 164). It is a smooth function of time that tells us what day of an invariant 365 day cycle the date t is. The function phase$(t)$ is modeled here as:

$$\text{phase}(t) = 365 \times \text{Beta}\left( \frac{t - \text{min·extent·day}(\text{year}(t))}{\text{max·extent·day}(\text{year}(t)+1) - \text{min·extent·day}(\text{year}(t))}; \beta(\text{year}(t)) \right) \tag{8}$$

$$\text{min·extent·day}(\text{year}(t)) \leq t \leq \text{max·extent·day}(\text{year}(t)) \tag{9}$$

where max·extent·day$(y)$ is the day of the year giving the maximum extent for year y and min·extent·day$(y)$) is the day of the
year giving the maximum extent for year $y$. Here Beta$(p; \beta), 0 \leq p \leq 1$, is the cumulative distribution function of a Beta$(\beta)$ random variable parametrized by $\beta = (\beta_1 > 0, \beta_2 > 0)$, and $\beta(y)$ is the parameter value specific to year y.

Here $a_P[s]$ is an invariant annual cycle for the extent (typically differing from $a_I[s]$). It is defined in an analogous way to the other invariant annual cycles as a cyclic cubic spline function of $s$ chosen to minimize the penalized square error:

$$\text{PSE}_{\lambda_P, \beta}(u) = \sum_{t=T_0}^{T} \left\{ \text{extent}(t) - u[\text{phase}(t; \beta(\text{year}(t)))] \right\}^2 + \lambda_P \int_{0}^{365} u''[s]^2 ds \qquad \lambda_P > 0, \quad \beta(\text{year}(t)) > 0 \tag{10}$$





where $\lambda_P$ is a smoothing parameter, chosen to balance the closeness of fit to the recorded values (the first term) with the smoothness of $u[s]$ (the second term). The minimization is also over the parameters $\{\beta_1(y) > 0, \beta_2(y) > 0\}_{y=1978}^{2018}$.

The phase adjusted annual cycle gives a different decomposition of the extent than the invariant annual cycle as it captures variation due to phase variation. It allocates that component of the variation in extent due to phase variation to the annual cycle rather than the residual term, $\alpha(t)$.

Surprisingly, the adjustment for phase shows even more improvement (63.9%) in the MSE than that for the amplitude adjusted annual cycle indicating that the phase contributes more to the variability of the annual cycle of SIE than the amplitude. Most studies of Antarctic sea ice variability focus on the amplitude at maximum and minimum extents but this analysis indicates that the phase (the timing of these extrema) is at least as important a contributor to the variability.

Finally, we can combine the amplitude and phase adjustment ideas to define an annual cycle that jointly adjusts for both. We

define the *amplitude and phase adjusted annual cycle* (APAC), $a_{AP}[s]$:

$$\text{extent}(t) = a_A[\text{phase}(t), \text{min·extent}(\text{year}(t)), \text{max·extent}(\text{year}(t))] + \alpha(t) \tag{11}$$

where $a_A$ and phase$(t)$ are defined as in equations Eq. (5) and (9). Note that they will be different functions as they are now jointly specified. As before, $a_A[s]$ is modeled as a cyclic cubic spline function of $s$ chosen to minimize the penalized square error:

$$\text{PSE}_{\lambda_{APAC}, \beta}(u) = \sum_{t=T_0}^{T} \left\{ \frac{\text{extent}(t) - \text{min·extent}(\text{year}(t))}{\text{max·extent}(\text{year}(t)) - \text{min·extent}(\text{year}(t))} - u[\text{phase}(t; \beta(\text{year}(t)))] \right\}^2$$

$$+ \lambda_A \int_0^{365} u''[s]^2 ds \qquad \lambda_{APAC} > 0 \tag{12}$$

where $\lambda_{APAC}$ is a smoothing parameter. The minimization is also over the parameters $\{\beta_1(y) > 0, \beta_2(y) > 0\}_{y=1978}^{2018}$. As for the other annual cycles (invariant, amplitude adjusted, phase adjusted), $\lambda_{APAC}$ is chosen by Generalized Cross Validation.

Fig. 2(b) compares the traditional annual cycle, with the recorded SIE for 2016, and the APAC produced by this model for

the same time period. The APAC is a much better fit to the recorded data and represents a large and significant improvement of 77.3% in MSE (Table 1). Table 1 clearly demonstrates the value of having multiple successive definitions of the annual cycle when decomposing the variation in the daily annual cycle of SIE.

The discussion above describes several different ways of defining the annual cycle of SIE. While an annual cycle adjusted for phase or amplitude only would not be the best estimate for the data, differences between them and the optimal estimated

annual cycle (i.e., APAC) could reveal sources of variability in the daily SIE.

### 3.2    Analyzing variation: Volatility, daily rate of change, anomalies, and trend

Estimating the annual cycle using our model allows us to calculate statistics that reveal the underlying variability in the daily SIE. Below we decompose the sea ice variation on the time dimension, moving up the temporal scale from the very short





**Table 1.** Comparison of the various proposed Annual Cycles in terms of how well they explain the variation in daily SIE. Values are given as percentages of mean squared error and the square root of mean squared error (RMSE).

| Model | Unexplained variation in SIE (RMSE) | Improvement in MSE compared to the Traditional |
|---|---|---|
| Overall mean (total variation) | 5.627 | - |
| Traditional annual cycle | 0.576 | 0% |
| Invariant annual cycle | 0.482 | 28.7% |
| Amplitude adjusted | 0.382 | 55.2% |
| Phase adjusted | 0.343 | 63.9% |
| Amplitude and Phase variation adjusted | 0.272 | 77.3% |

term, the instantaneous variation, to the day-to-day variation, followed by the interannual variation and finally the multidecadal variation - the trend.

The recorded sea ice extent will deviate from the true sea ice extent. This may be due to some combination of weather, sea ice motion, artifacts of the satellite algorithm used for retrieval, and the electromagnetic spectrum across which the device/satellite is measuring, among other things. To represent this, we write the recorded SIE, $\text{SIE}(t)$, as:

$$\text{SIE}(t) = \text{extent}(t) + \epsilon(t) = a_A[\text{phase}(t), \text{min·extent}(\text{year}(t)), \text{max·extent}(\text{year}(t))] + \alpha(t) + \epsilon(t) \tag{13}$$

The recorded SIE on any given day is then the sum of a number of components of variation – the annual cycle for that day, the yearly variation (anomaly) from the annual cycle, and a residual term, $\epsilon(t)$. These are now discussed.

### 3.2.1 Volatility of the recorded sea ice extent

Here we introduce the term volatility to describe the instantaneous variation (or precision) in the recorded SIE as an approximation for the extent. Such variation may be due to ephemeral effects like those mentioned above.

Normally the standard deviation of the residual, $\epsilon(t)$ in Eq. (13) is represented as a constant over time. Here, however, we allow it to vary, explicitly representing it as a time varying term/component. The *volatility* is therefore defined as the time series formed by the standard deviation of $\epsilon(t), t = T_0, \ldots, T$. It is a quantification of ephemeral effects. Effectively it shows the size and timing of the variability associated with factors like instrument error or noise in the recorded SIE.

To model the volatility, we specify a generalized autoregressive conditional heteroskedasticity (GARCH) model (Bollerslev, 1986) for the residual $\epsilon(t)$. The residual is split into a time-dependent standard deviation $\sigma(t)$ representing the volatility and a series $z(t) \sim N(0,1)$:

$$\epsilon(t) = \sigma(t)z(t)$$

Explicitly, the (squared) volatility is modeled as a weighted average of the past anomalies and (squared) volatilities:

$$\sigma^2(t) = \omega + \sum_{i=1}^{p} \eta_i \epsilon(t-i) + \sum_{i=1}^{q} \psi_i \sigma^2(t-i)$$



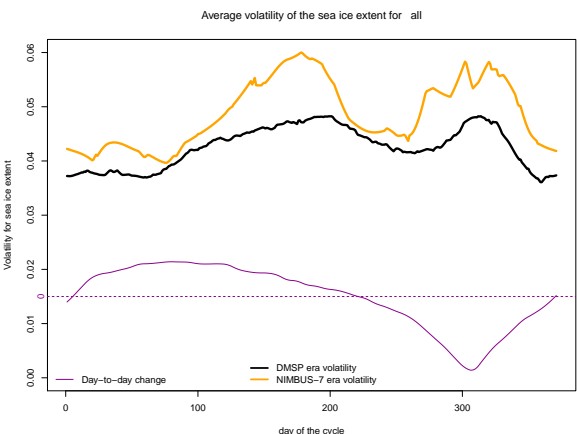

**Figure 3.** Volatility of the recorded SIE for the NIMBUS-7 era (26 October 1978 through 20 August 1987) and the DMSP era (21 August 1987 thru the present). It is averaged over each day-of-the-cycle in these eras. The units are million km$^2$. The purple curve is the day-to-day change on SIE from Fig. 4

where the parameters $\eta_i$ and $\phi_i$ represent dependency on the past residuals and volatilities, while the parameter $\omega$ represents a trend in volatility. The purpose of the dependency on past volatilities is to better represent periods of high or low volatility. We also specify an Autoregressive Moving Average (ARMA) model for $\alpha(t)$ (Box and Jenkins, 1976; Hipel and McLeod, 1994) with $\epsilon(t)$ as the (time-dependent) error term. The model parameters were fit using maximum likelihood. The Bayesian Information Criterion (BIC) was used to select the model order (Ghalanos, 2019). The model orders were $p = 2$ and $q = 2$ (i.e., GARCH(2,2)) and auto-regressive moving average, ARMA(1,1), for the anomaly model. All models were fit using the R package `rugarch` (Ghalanos, 2019).

Fig. 3 plots the average volatility in SIE, separating it into the two periods of time when different sensors were retrieving the data. It is clear that the volatility is larger in the data recorded by the NIMBUS-7 sensor (orange) than by the DMSP (black) especially at times of maximum SIE. This is a difference that must be taken into consideration when using this variable (volatility) across the whole time-series. That said, there are some important similarities. Volatility is least at SIE minimum, larger at SIE maximum and largest late in the cycle when the ice is experiencing its largest rate of retreat. This latter characteristic is discussed below. The values from the DMSP era show that the volatility ranges from approximately 40 - 50K km$^2$. These are relatively small values compared to the total SIE but quite large compared to the typical grid cell size. The fact that the volatility is not constant over the cycle may be exploited to get a better understanding of contributors to overall variability in SIE.



### 3.2.2 Daily rate of change

It is useful to know the daily rate of change of SIE because it gives insight into the daily timing of growth (advance) and melt (retreat) of the sea ice. It is also an expression of the phase of the annual cycle. Contemporary trends in Antarctic sea ice are shown to be linked to the changes in the timing (phase) of advance and retreat (e.g., Stammerjohn et al., 2008). Note that the annual cycles have been defined as continuous in day. Hence, we can quantify the rate of change of total Antarctic SIE by the derivative of an annual cycle shape function, $a[s]$. The precise definition of the rate of change differs by the choice of annual cycle to use. As an example, the rate of change for both the traditional and invariant annual cycles is plotted in Fig. 4 which shows the day to day changes in the SIE over the 365 day cycle. As might be expected, the overall pattern of the traditional (orange line) and invariant annual cycles (black line) are quite similar to each other. Both cycles show that the rates of growth and melt are variable over the cycle. However, compared to that of the invariant, the day to day change in the traditional annual cycle is quite variable, making it difficult, if not impossible, to make precise statements about the timing of ice growth and decay. Therefore, the following comments are based on the day to day change in the invariant annual cycle. The SIE minimum (day 0, Julian Day 46) is coincident with the minimum growth rate. The ice advances, reaching maximum growth rate by day 81 maintaining this maximum growth rate for approximately 40 days before slowing to a minimum growth rate by day 225 (late September) of the cycle. Sea ice retreat begins at approximately day 225 and occurs quite rapidly compared to advance, reaching a maximum rate at day 308 (late December) before slowing to a stop at day 365 (Julian day 46 or mid February). The rates of advance and retreat of the ice are not constant over the annual cycle. The maximum rate of retreat of the ice is more than twice the maximum rate of advance. Fig. 4 illustrates and defines more precisely a key characteristic of the Antarctic annual cycle, that is, its asymmetry. The ice grows (advances) steadily over a much longer period than its decay (retreat) It has been suggested that this asymmetry in the annual cycle is a result of the influence of the semi-annual oscillation (SAO) of the Antarctic circumpolar trough (Enomoto and Ohmura, 1990; Watkins and Simmonds, 1999) and an open water (ice)–albedo feedback with the latter being the main driver for the rapid retreat of sea ice (Ohshima and Nihashi, 2005). Recent modeling studies (Kusahara et al., 2018) suggest that ice advance is due chiefly to thermodynamic processes (except in the Ross Sea) while ice retreat is largely wind driven (or dynamic). Our study provides more precise information on the timing of advance/retreat and on the length of two major stages of the ice cycle - ice growth - 7.5 months, 225 days; ice retreat-140 days, 4.5 months - than can be obtained from monthly averaged data. This is significant because much of the variation in contemporary Antarctic SIE has been occurring at sub-monthly scales.

Taken together, the daily rate of change and the volatility (Fig. 3-4) show, (1) The timing of lowest volatility may be related to the fact that there is relatively little ice at minimum; (2) During the period when ice is advancing most swiftly, the volatility is low, responding to constant large scale forcing; (3) During the period of slowing growth and maximum extent, volatility is high, perhaps due to the more frequent occurrence of storms during winter (Simmonds and Keay, 2000) causing fluctuations at the sea ice edge rather than within the pack where the sea ice concentration is at or close to 100%; (4) Volatility begins to decrease as the sea ice retreats; but (5) increases to its maximum value when the rate of retreat is largest. The late peak in volatility may be due to the dynamic nature of the retreat. Anecdotally, the sea ice extent anomalies of note tend to occur during

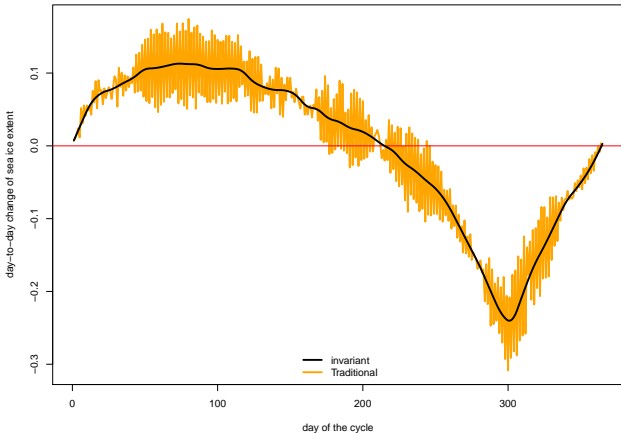

**Figure 4.** Day-to-day change in the annual cycle of sea ice extent for the traditional (orange) and invariant (black) annual cycles. The horizontal axis is the day of the cycle and the vertical axis is change in sea ice extent in millions of km$^2$.

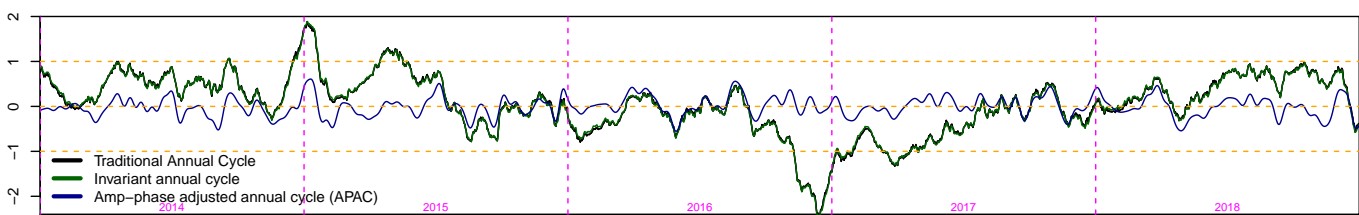

**Figure 5.** Comparison of anomalies from three Annual Cycle estimates for 2014-2018 - the raw anomaly from the traditional annual cycle (black), the estimated anomaly from the invariant annual cycle (green), the estimated anomalies from the Amplitude-Phase adjusted annual cycle (blue). The vertical axis is the anomaly in millions of km$^2$.

the sea ice maximum and the period immediately following (Turner et al., 2017; Schlosser et al., 2018). The statistics examined
here are suggesting that these anomalies are probably associated more strongly with dynamic forcing than thermodynamic.

### 3.2.3   Anomalies

The detection and analysis of anomalies (deviations from the annual cycle) is essential to the understanding of contributors to
variability. Here we discuss three different but related types of anomalies. First there is the *true anomaly*, represented by $\alpha(t)$
in Eqs. (1), (11) and (13). This is the difference between the true SIE and the annual cycle, however defined. The true anomaly
is the preferred anomaly but is unobtainable because of imprecision in measuring and retrieving the sea ice data. Second there
is the *raw anomaly*, the difference between the observed (recorded) SIE and the annual cycle. Here we focus on a statistical
estimate of the true anomaly, $\alpha(t)$, which we denote by $\hat{\alpha}(t)$. The estimate is preferable to the raw anomaly as it adjusts for the
volatility and should be closer to the true anomaly than the raw anomaly.





We estimate the true anomaly by using Eq. (13), rewriting it as

$$\hat{\alpha}(t) = \text{SIE}(t) - \hat{a}_A[\hat{\text{phase}}(t), \text{min·extent}(\text{year}(t)), \text{max·extent}(\text{year}(t))] - \hat{\epsilon}(t) \tag{14}$$


We use the estimate of the APAC and compute $\hat{\epsilon}(t)$ from the GARCH model for the residual $\epsilon(t)$ from Sect. 3.2.1. The estimated anomaly is quite close to the recorded anomaly as $\hat{\epsilon}(t)$ is small in magnitude (See Fig. 3 and 7).

Fig. 5 plots three types of anomalies: the raw anomaly from the traditional annual cycle, and the estimated anomalies from the invariant and APAC. These show the last five years of the 42 years of satellite-observed data, 2014-2018. The anomalies of

the three annual cycles are similar in sign however, those for the APAC tend to be smaller. The similarity in sign is expected and the smaller size of the APAC anomalies arises because the APAC is a much better fit to the recorded data. The anomalies for the traditional and invariant annual cycles are not significantly different from each other in size. This is expected given the small difference between the two shown in Fig. 2. We can clearly see the large negative anomaly in SIE at the end of 2016. The negative anomaly is larger in the traditional and invariant annual cycles than in the APAC, demonstrating that the APAC is

a better fit to the recorded SIE therefore the anomaly is expected to be smaller.

### 3.2.4 Trend

The trends in SIE for both the Arctic and Antarctic have been the subject of much study. Most studies assume a linear trend and employ a linear model of the monthly data to estimate those trends (e.g., Parkinson and Cavalieri, 2012). Instead, we remove this assumption of linearity and model the trend in the daily data as a thin plate regression spline function of time (Wood,

2003). We added a term to our model for the SIE representing this curvilinear trend and jointly estimate it by minimizing the PSE (penalized square error):

$$\text{PSE}_{\lambda_I, \lambda_{\text{trend}}}(a, \text{trend}) = \sum_{t=T_0}^{T} \{\text{extent}(t) - \text{trend}(t) - a[\text{doy}(t)]\}^2$$
$$+ \quad \lambda_I \int_0^{365} a''[s]^2 ds \quad + \quad \lambda_{\text{trend}} \int_{T_0}^{T} \text{trend}''[t]^2 dt \qquad \lambda_I > 0, \quad \lambda_{\text{trend}} > 0 \tag{15}$$

where $\text{trend}''[t]$ is the 2nd derivative of $\text{trend}[t]$ at time $t$ and $\lambda_{\text{trend}}$ is a smoothing parameter specific to the trend and is chosen

to balance the closeness of fit to the recorded values using Generalized Cross-validation (Wood, 2004).

The curvilinear trend in SIE for 1979-2015 and 1979-2018 derived using this method is illustrated in Fig. 6 along with the linear estimates of the trend. The latter assumes the same model as Eq. (15) except it constrains $\text{trend}(t)$ to be linear. While there is a small positive linear trend, as has been reported in the literature (Parkinson and Cavalieri, 2012; Turner et al., 2015a, e.g.,), Fig. 6 shows that there is strong non-linearity in the trend. There are strong decadal differences. For example, in the 1980s

the trend was largely negative, while from 1990 to the mid 2000s, there were a number of short-term fluctuations with opposing signs. It seems clear from Fig. 6 that the reported positive trend in total Antarctic SIE is due largely to the positive trend that began at the end of the first decade of the 21st century and continued until 2016. The anomalously low SIE experienced since

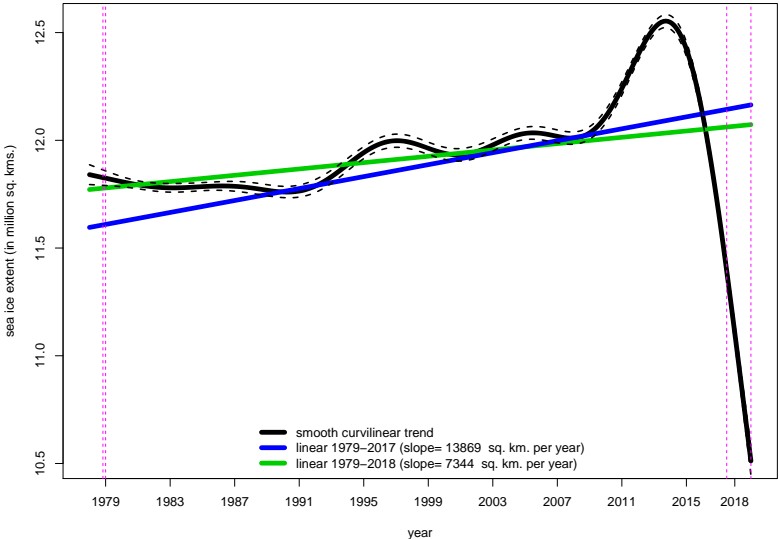

**Figure 6.** Three estimates of the trend in the recorded SIE represented in terms of the amount of SIE associated with the change. The blue is the linear trend estimated for data from 1 January 1979 through 31 December 2017. The green is the linear trend estimated for data from 1 January 1979 through 31 December 2018. The black is the curvilinear trend estimated for data from 26 October 1978 through 31 December 2019.

2016 had the effect of reducing the slope of the linear trend by almost 50% - from 13860 km$^2$ per year to 6068 km$^2$ per year. The nonlinearity of the daily SIE trend in this analysis is consistent with that discussed by Simpkins et al. (2013) in their

analysis of changes in the magnitudes of the sea ice trends in the Ross and Bellingshausen Seas. We note also that use of the daily data adjusted for amplitude and phase potentially allows a better estimate of the trend than monthly averaged values.

Even within the context of nonlinearity, the anomalously low SIE represents a dramatic negative adjustment to Antarctic SIE (Schlosser et al., 2018; Parkinson, 2019), prompting questions about whether or not this represents a change in state, instead of a fluctuation due to natural variability. The current length of record does not allow much more than speculation. However,

we can decompose the annual cycle of 2016 into the various components of variation that we have identified in this paper. This is illustrated in Fig. 7. The daily values of the components are plotted against the anomaly in SIE, showing how much they contributed to the SIE anomaly. The decomposition is sequential with the amplitude component extracted before the phase component.

The decomposition shows that the curvilinear trend (green) for 2016 is small and positive early in the cycle becoming

strongly negative later in the year and making a large contribution to the negative anomaly during this time of rapid change identified in Fig. 6. The raw anomaly (black), the difference between the recorded SIE and the APAC - the anomaly which includes the volatility - and its smoothed version, the estimated anomaly (orange), is small and did not make a consistent contribution to the anomaly in SIE over the year. Smoothing removed the "noise" which might be due to instrumentation and leaving behind a truer variation between the recorded SIE and the expected SIE (i.e the APAC). The amplitude (blue) made





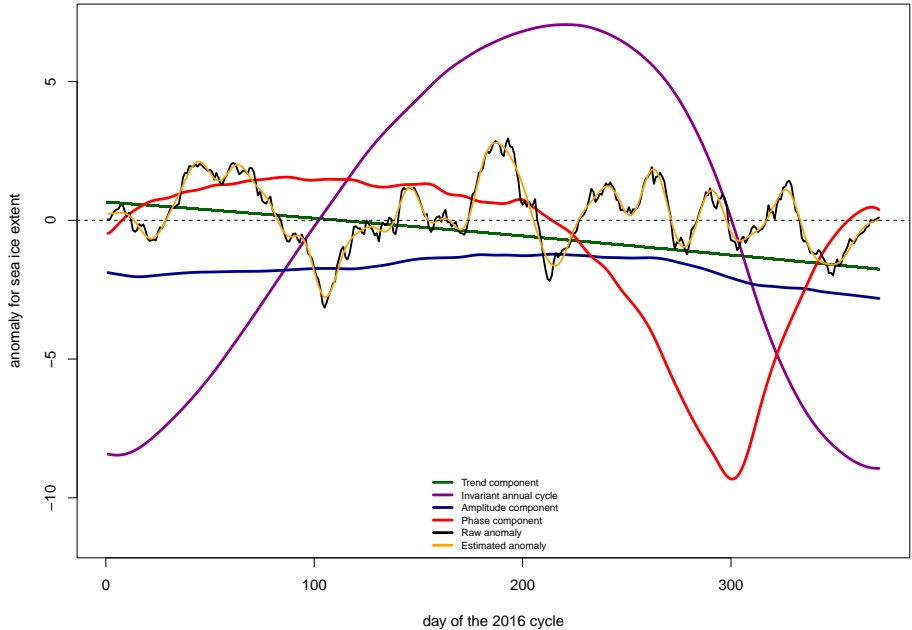

**Figure 7.** Decomposition of the sea ice extent during 2016 into various components of its variation, including separate amplitude and phase components. Day of cycle is on the horizontal axis - day 0 is Julian day 52. Anomalous SIE in millions of $km^2$ is on the vertical axis.

a small but consistently negative contribution to the anomalously low SIE in 2016. Interestingly, the main contributor to the anomalous SIE was the phase. That is, the phase contributed to a small positive anomaly during the growth stage of the cycle (the growth was slightly ahead of phase) and a strongly negative anomaly during retreat indicating that the timing of retreat of the ice was earlier than normal and the ice retreated faster than normal. The sum of these components (including the invariant annual cycle (magenta) is the recorded SIE for 2016. The decomposition shows that the difference between the recorded SIE and the traditional and invariant cycles seen in Fig. 2(a) is mostly due to phase.

## 4   Conclusions

Variability in the annual cycle of Antarctic sea ice extent is dominated by the amplitude and phase of the cycle. In this study, we examined in detail the variability in the annual cycle of total Antarctic sea ice extent (SIE) at timescales ranging from the instantaneous, the day-to-day change, the interannual, to the multidecadal trend, thus offering a complete picture of the temporal variation of the sea ice. To facilitate this analysis, we developed first a statistical and mathematical model of the annual cycle in which the amplitude and phase, the two major contributors to its variability, are allowed to vary. This is contrary to traditional methods which restrict the variation of amplitude and phase thus limiting their contribution to the variability. We define a number of complementary annual cycles – the invariant, which is an optimally smoothed annual cycle with no adjustments for phase or amplitude, an annual cycle which adjusted for phase only, another adjusted for amplitude only





and one that is adjusted for phase and amplitude (APAC). Each of these annual cycles represent clear conceptual and statistical improvements over the traditional method of calculating the annual cycle, with the APAC showing the most improvement. We propose the APAC as a substitution for the traditional method. However, the differences between the other annual cycles and the APAC reveal sources of variability in the daily SIE. For example, comparing the annual cycles adjusted for phase only and amplitude only revealed that the phase contributes more to the variability in the annual cycle than the amplitude.

The timescales into which the variability of SIE was decomposed allow useful interpretations of the factors that give rise to the variability. Using the volatility, the volatility defined and described here for the first time, we show how much of the total SIE is due to ephemeral effects. We also show how those ephemeral effects vary over the annual cycle and in the process, we note that there are differences in the volatility (and hence uncertainty) that arise because of sensor type. The daily rate of change in SIE allows a precise definition of the timing and rate of advance and retreat of the sea ice, a quality that is very 330 important given that much of the contemporary variability in Antarctic sea ice occurs at sub-monthly scales. Combination of the information given by the volatility and daily rate of change suggests that the volatility is lowest when the sea ice is at minimum and highest during the time of maximum rate of retreat. Given that the rapid rate of retreat of the ice has been associated with dynamic processes this suggests that the peak in volatility at the end of the cycle is due to ephemeral effects associated with dynamic forcing.

To look at the interannual timescale, we defined/estimated several different but related anomalies, measures of deviation from the annual cycle, that may be used to evaluate the contributions to Antarctic sea ice variability from sources (local, oceanic, and atmospheric) other than the large scale sources that control cyclical, amplitude and phase changes. These show that our proposed annual cycle, the APAC, is a better fit to the recorded SIE.

We established that the trend in daily, total Antarctic SIE over time is strongly nonlinear and that the linear estimates are 340 weak and dependent on a positive trend that began in 2011 and ended in 2016. Interestingly, our decomposition of the annual cycle of 2016 into the components of variation defined in this paper shows that the main contributor to the anomalous SIE was the phase. That is, the anomalously low SIE was due mainly to the fact that the retreat began earlier than normal and was faster than normal. The amplitude made a much smaller negative contribution that did not vary much over the year.

We used the daily, total Antarctic SIE in this analysis. However, sea ice variability around Antarctica is strongly regional, 345 and the annual cycle of these regions are markedly different from each other and changing. The model-estimated annual cycles and the timescale decomposition presented here will facilitate examination of the regional variability of Antarctic sea ice. Finally, although our method was developed on Antarctic SIE, this decomposition methodology is applicable to a wide range of climatic variables (e.g temperature, Arctic sea ice extent) that experience an annual cycle.

*Code and data availability.* The data used to generate the sea ice extent are freely available from the National Snow and Ice Data Center 350 (NSIDC) (Meier et al., 2017; Peng et al., 2013). Upon publication, the R language (R Core Team, 2019) software used to produce the analysis in this paper will be make available on an open source repository on GitHub (Handcock et al., 2019). This means all figures and numbers in this paper can be reproduced. In addition, the code will be submitted for peer review and as a contribution to the Antarctic/Southern Ocean





rOpenSci project (Raymond and Sumner, 2018) that is a collaboration between the Scientific Committee on Antarctic Research (SCAR)and rOpenSci.

*Author contributions.* M.N.R conceived the idea for this study. M.N.R and M.S.H equally developed the statistical methodology and analyzed the data. M.S.H. wrote the software and process of the data. Both authors assisted in writing the editing the manuscript.

*Competing interests.* The authors declare that they have no competing interests.

*Acknowledgements.* This work was supported by the National Science Foundation under the Office of Polar Programs under grant NSF-OPP-1745089.



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
