# Peer review of "Modeling the annual cycle of daily Antarctic sea ice extent"

_The Cryosphere, 2019_

## Referee Comment (RC1) · Walter Meier (Referee) · 16 Dec 2019

Summary:

This paper analyzes the seasonal cycle and interannual variability of Antarctic sea ice extent (SIE) using various statistically approaches. Different annual cycles are defined based on amplitude and phase. Variation in phase is found to explain more of the SIE variability, but combining both amplitude and phase explains substantially more variation than traditional methods. The approach shows that the low SIE extremes in 2016 were due mainly to a shift in phase.

General Comment:

This paper makes an interesting contribution to Antarctic SIE analyses. Two lingering

questions with Antarctic SIE is the small positive trend that has been seen over the long-term satellite timeseries and the whiplash in recent years going from record highs to record lows within just a couple years. The analysis presented in this paper is unique for SIE and the paper brings to light many relevant characteristics of the SIE timeseries that provide insights into both short-term and long-term variability. For example, the idea that phase plays a more important role in the seasonal cycle than amplitude is revealing and seemingly important in better understanding the character and variability of Antarctic SIE. I have a few minor comments on various aspects. I recommend acceptance after minor revision to address these.

Specific Comments (by line number):

1: "troughing", while technically correct, reads awkwardly to me. Why not just say "reaches its minimum"? This occurs in a few other places in the text.

23: "peaking in September. . .and troughing in late February. . .on average." Though you say on average, which is accurate, that masks a lot of variability. The minimum does sometimes occur in October and ranges from early Sept (even late Aug one year) through early Oct. The maximum can occur in March and ranges over ∼3 weeks. It might be worth providing a range along with the average to give a better sense of the variability, which as is shown later in the paper is important

29: Some of the day-to-day variation is also due to land-spillover (coastal effect of mixed land/water grid cells). It's not as variable as weather or changes in the ice cover, but I think it is important enough to warrant mention. (This is less of an issue in Antarctica because of the land ice along the coast, but still worth noting I think.)

61-63: The data reference is a little confusing. You say use the SMMR-SSMI-SSMIS Bootstrap Version 3 product, but reference Comiso (2017), which is the correct reference. But you also reference Peng et al. (2013), and Meier et al. (2017), which refers to the NSIDC/NOAA Climate Data Record product. I understand the confusion here because the NSIDC/NOAA CDR does include the Bootstrap V3 concentrations

within the product. My assumption is that you used the Bootstrap V3 field within the NOAA/NSIDC CDR. So I think all three references are warranted, but this could be more clearly explained, e.g., "We used the Bootstrap Version 3 concentration fields (Comiso, 2017) from the "NOAA/NSIDC Climate Data Record of Passive Microwave Sea Ice Concentration, Version 3" (Peng et al., 2013; Meier et al., 2017)." Or something like that.

66: You note that there are a number of days with no observations (in addition to the every-other-day SMMR). But one of those gaps is quite significant, with no data between early December 1987 and mid-January 1988. This is worth noting because it is unique in the record in terms of the length of the gap. Did you fill this in at all or leave the gap? Since the method doesn't require complete data, I assume not, but that should be made clear.

69: Day 0 is the minimum day of the year and then you just plot the next 364 days after that for each year. But of course, the date of the minimum differs from year to year. So, it seems like some years could have a gap – if the minimum of one year occurs before the minimum of the next year (i.e., >365 days between minimums) – where some data is not plotted, or conversely, you could have some data duplicated – if the minimum of one year occurs later than the minimum of the next year (i.e., <365 days between minimums). Is this correct? Are these "missed" or "duplicated" accounted for in any way? Or does that potentially skew results at all?

Figure 3: A few suggestions. First, the Day-to-day change is in Figure 4 (as noted in the Figure 3 caption). It seems like it is discussed in the context of Figure 4. So, is it necessary to include that line in Figure 3? Simpler is always better in my view, so one less line is helpful. And that would allow the y-axis to cover a smaller interval, which would more clearly show the variability lines. One thing that would be useful would be to label the max and min days (e.g., text with an arrow pointing to each). The day-to-day change does provide this, but it may not be immediately clear that the max occurs when the change turns from positive to negative. So, I think labeling would be helpful

even if the day-to-day line is kept (but if labeling is included, then that line isn't really needed). The fonts on these figures are quite small – in the final version, they should be much better. Also, while the units are noted in the caption, it my view it is always better to include them with the axis labels. Similarly, for Figure 4.

Figure 4: There is an interesting feature in the traditional (orange) plot right around day 200, where the curve is less dense. All the other places have thin lines, highly varying day to day. But around day 200, there seems to be a period where the line just peaks and then declines over several days. Is that related to anything? Or is that just a quirk in the data, or just an optical illusion?

206-214: Why is the volatility higher for SMMR than for DMSP? Is it simply the every-other-day sampling? But there could also be an effect due to the sensor resolution (sensor footprint), which is actually smaller (higher resolution) in SMMR. I'm curious if the volatility of DMSP would match SMMR if every-other-day values were used from DMSP? Another, smaller aspect, is whether volatility changes from SSMI to SSMIS? If it's simply the temporal sampling, then I would expect there wouldn't be a change. But if there is a resolution component, then there might be a small effect since the sensor footprints are slightly different. While I think looking at that could be interesting, I guess it's not the main focus of the paper, so I can see not doing that. However, I think it is worth at least noting that the differences in volatility are due to temporal sampling (and maybe some resolution effect?), just to make that clear.

244: There is also more volatility at/near the maximum because there is more ice edge to vary. At the maximum, the perimeter of the ice cover is also at or near its maximum, which allows more areas to be affected by winds, currents, storms, etc.

Figure 5: What are the anomalies relative to – i.e., what is the base period? Likewise, for earlier figures, the y-axis should be labels with units.

281-291: I understand the rationale for using the daily values over monthly values, but the advantage of monthly values is that you capture roughly the same period in the

cycle – so you can look at trends near the maximum or near the minimum, which can be quite different than over a full year. But I also wonder is something is lost? – you're taking something with a big annual cycle and then just fitting trend lines through the entire ~40 years. Would it make sense to do a Figure 6 for the max and min? Perhaps using the amplitude and phase adjusted? Also, how does the curvilinear trend handle the endpoints – i.e., how does it calculate a trend from the beginning? In other words, how does the function (Eq. 15) calculate a smooth trend at the beginning of the time series? I assume that there is an endpoint fitting/smoothing, which may be in the equation. But some plain English explanation would be helpful as well.

Figure 6: What are the thin pink dashed lines? Are these just the beginning and end dates of the two periods? And the dashed line around the curvilinear trend?

288: The trend standard deviation (+/-) values should be included with the linear trend and maybe the trend significance.

---

## Referee Comment (RC2) · Anonymous Referee #2 · 29 Jan 2020

General Comments

The paper proposes a statistically based framework for investigating the annual cycle in Antarctic sea ice extent (SIE). The paper delves into the different drivers of variation in the annual cycle, with a focus on the amplitude and the phase. Many researchers (myself included) who are interested in Antarctic Sea Ice and its changes over the satellite era, have or are, pondering over the drivers of change in SIE. I would recommend that the paper is published with minor revisions, much of which from my personal perspective, focus on the accessibility of the paper.

Main Comments - Text

This work will be of great interest to the entire community interested in Antarctic sea

ice, from researchers focused on SIE through to biologists, glaciologists and those involved in the operational aspects of Antarctic science. To that end, I would ask the authors to consider whether the paper can be improved in terms of its accessibility. Reviewing this, I am required to read every line, and as such, I found that I needed several "sittings" to complete my first read through. I would deal with a section, but I would then need a break of several hours before I felt ready to tackle the next section. The nature of this work, the detail and effort that has gone into developing the method, does to an extent require this level of complexity, but my fear is that it might push other readers away, meaning they miss the crucial detail within the methods. To this end, I have a few suggestions for improving the accessibility:

A) Introduction

Lines 23-37 quickly bounce from introducing the annual cycle and its year on year variability to delving into issues surrounding satellite retrieval then into complexities regarding the duration of the record itself. I think this should be split up slightly, particularly the focus on the components of the annual cycle in Lines 23-26. Amplitude and phase are crucial throughout much of the paper and I think it would be of great value for the authors to spend some time here, defining them and why they are important. I would also remove the brief mention of the regional variations in the annual cycle, the body of this work will focus on pan-Antarctic SIE, and while there are interesting developments to this work looking at specific regions, it is fairly, not covered in the majority of this paper.

Once the components of the annual cycle are defined, it will be then easier to outline why they are affected by the current retrieval methods and the duration of the record. This then allows a better set up for why this work is necessary.

I would delete Lines 38-41 (up to and including "climate data"). No specific examples of other annual climate cycle issues are outlined, nor does it seem that the methods used here are applied elsewhere (if they are, then please state this with the example
more clearly). Removing these lines would allow better flow from outlining the issues into your over-arching aims.

Line 51 is a good example of the general accessibility of the text; "the mathematical and stochastic representation of proximate forces" is potentially obtuse. The (very) similar sentence that follows from Line 52-53 is far more accessible to a less statistically minded reader.

B) Methodology and Results

Each process is defined with respect to the model, previous models and the statistical analysis involved. What I think would benefit this section is for an introductory section at the beginning of Section 3 that defines each term for: • Annual Cycle • Invariant Annual Cycle • Amplitude and Phase Adjusted Annual Cycle

Highlighting their importance to understanding the cycles and the changes. The final line for each of these would point to the following section where they are defined and their results discussed. By creating this section, a reader can easily refresh themselves as to what is each of these components, as that is crucial to understanding the results.

The section would also benefit from the results for each cycle being a new paragraph to ensure that they stand out, currently in most of the sections it runs straight from the methodology behind the cycle into its result. This runs the risk of the result being missed by readers.

Main Comments - Figures

Figure 1

The smooth annual cycle as a blue line is not distinguishable as blue. I would change the colour so that it can be resolved by the reader, even if that was black, which is what the line mostly appears to me currently (both on screen and in print). Figure 1 also sets a convention that Day 0 is the start of the cycle and Day 365 is the end of the cycle. However, the annual cycle is rarely exactly 365 days, it is "on average" but year on year

it is not. How does this impact on both the figures and also the paper analysis? This should be addressed in the body text of the paper.

Figure 2

Please label the panels as A and B. In general both panels are too small to properly resolve the detail within the image, particularly from the lines that overlap. In A, the green and black are nearly indistinguishable to my eye.

Figure 3

The title over the figure appears to be incomplete. Given a title doesn't appear on the other figures is this an error for it to still remain?

Figure 5 and Figure 6

I'm not a fan of green, blue and black lines on the same figure, they are very hard to resolve, particularly the green and black which heavily overlap. The same issue also applied to Figure 6.

Figure 7

In a similar vein to Figures 5 & 6, the use of maroon and red on the same figure makes the figure harder to interpret. The figure legend also needs to be significantly bigger to make it more readable.

Specific (Line-by-Line) Comments

Line 16-17: The ordering of the references is ad-hoc, neither in publication date order or alphabetical order based on first author initial. In Line 20, they are ordered by date of publication, please adjust Line 16 to match.

Line 23: In the abstract (Line 2) you state that the SIE minima is in March; here in Line 23 you state it is late February. Please ensure consistency between these two dates. (See also Line 230-231 comments and Figure 1 comments)

Line 24: I would suggest altering this line to read: "The growth from minimum (trough) to maximum (peak) is slower than the retreat from maximum to minimum". Your use of trough refers to the minima, not the slope getting there, so I think defining both terms straight away fits better.

Line 39 (and others): Some sections of the paper, such as here use an example reference (Stine et al., 2009) is followed by "e.g.,". Elsewhere in the paper it precedes the reference (i.e. Line 29 "(e.g. Comiso and Steffen, 2001)"). I would prefer the Line 29 example, but either way is acceptable as long as it is consistent throughout the paper.

Lines 58-59: The outline of the sections is not consistent with the body text. Results are in Section 3 alongside the Methods and Conclusions are in Section 4.

Line 81: T = 2019. I was unsure if this should be T = 2019.000; T = 2019.999 or something in between. Could this be clarified.

Line 89: I would appreciate a short (1 or 2 lines) explaining why you chose not to consider this data further. To raise it as a possible way to increase your temporal range and then simply dismiss it feels incomplete to me.

Line 230-231: Here the minima is listed a mid-February, which is different to the use of March (abstract) and late-February (i.e. Line 23). The consistency of the definition of the minima throughout the paper would be useful (albeit tricky due to the variability in the occurrence of the minima; as mentioned in the comments on Figure 1, this could do with being addressed earlier in the paper)

Lines 236-239: Ice budget analysis work (i.e. Holland & Kwok, 2012; Holland, 2014; Pope et al., 2017; Holmes et al., 2019) has indicated that the surface winds play a role (advection and divergence terms within the budget) throughout the whole of West Antarctica and into the Weddell Sea in both the advance and the retreat of sea ice. I would mention this work here with respect to the drivers of the advance and retreat of sea ice in addition to the modelling study mentioned.

Line 239-240: the sudden use of months (when most of the time dimension to date in the paper has been in days) is unnecessary and makes for awkward flow. I would remove 7.5 months and 4.5 months from these lines, as it would clean up the sentence which has to many "-" in it, making for a poor flow.

Line 349: The authors should be praised for being so diligent in making their code accessible and open to peer review.

---

## Author Comment (AC1) · 18 Mar 2020

**General Comments**

The paper proposes a statistically based framework for investigating the annual cycle in Antarctic sea ice extent (SIE). The paper delves into the different drivers of variation in the annual cycle, with a focus on the amplitude and the phase. Many researchers (myself included) who are interested in Antarctic Sea Ice and its changes over the satellite era, have or are, pondering over the drivers of change in SIE. I would recommend that the paper is published with minor revisions, much of which from my personal perspective, focus on the accessibility of the paper.

**Main Comments - Text**

This work will be of great interest to the entire community interested in Antarctic sea ice, from researchers focused on SIE through to biologists, glaciologists and those involved in the operational aspects of Antarctic science. To that end, I would ask the authors to consider whether the paper can be improved in terms of its accessibility. Reviewing this, I am required to read every line, and as such, I found that I needed several "sittings" to complete my first read through. I would deal with a section, but I would then need a break of several hours before I felt ready to tackle the next section. The nature of this work, the detail and effort that has gone into developing the method, does to an extent require this level of complexity, but my fear is that it might push other readers away, meaning they miss the crucial detail within the methods. To this end, I have a few suggestions for improving the accessibility:

**Introduction**

Lines 23-37 quickly bounce from introducing the annual cycle and its year on year variability to delving into issues surrounding

satellite retrieval then into complexities regarding the duration of the record itself. I think this should be split up slightly, particularly the focus on the components of the annual cycle in Lines 23-26. Amplitude and phase are crucial throughout much of the paper and I think it would be of great value for the authors to spend some time here, defining them and why they are important. I would also remove the brief mention of the regional variations in the annual cycle, the body of this work will focus on pan-Antarctic SIE, and while there are interesting developments to this work looking at specific regions, it is fairly, not covered in the majority of this paper.

*Authors' response:*

We agree.

*Authors' changes in the manuscript:*

The suggested modifications to this part of the Introduction will be adopted.

Once the components of the annual cycle are defined, it will be then easier to outline why they are affected by the current retrieval methods and the duration of the record. This then allows a better set up for why this work is necessary.

I would delete Lines 38-41 (up to and including "climate data"). No specific examples of other annual climate cycle issues are outlined, nor does it seem that the methods used here are applied elsewhere (if they are, then please state this with the example more clearly). Removing these lines would allow better flow from outlining the issues into your over-arching aims.

*Authors' response:*

We understand the referee's point. We will remove the lines 38-41.

*Authors' changes in the manuscript:*

Lines 38 - 41 have been removed.

Line 51 is a good example of the general accessibility of the text; "the mathematical and stochastic representation of proximate forces" is potentially obtuse. The (very) similar sentence that follows from Line 52-53 is far more accessible to a less statistically minded reader.

*Authors' response:*

We will rewrite this section to make it more accessible to the reader.

*Authors' changes in the manuscript:*

We will change the sentences starting this paragraph to:

"We begin by presenting a stochastic model for the sea ice extent that allows the annual cycle to be defined in flexible ways. This model can represent the real variability in SIE and reduces the contribution from the ephemeral effects described above. The model can account for the fact that the ice maximum is not achieved on the same day of the ice cycle each year. It also recognizes that the length of the ice cycle will vary and that the timing of advance and retreat of the ice varies from year to year. This means that the annual cycle is not constrained to be a fixed cyclical pattern rather, it is a pattern that allows both temporal dilation and contraction as well as amplitude modulation."

Methodology and Results

Each process is defined with respect to the model, previous models and the statistical analysis involved. What I think would benefit this section is for an introductory section at the beginning of Section 3 that defines each term for:   c´ Annual Cycle   c´ Invariant Annual Cycle   c´ Amplitude and Phase Adjusted Annual Cycle

Highlighting their importance to understanding the cycles and the changes. The final line for each of these would point to the following section where they are defined and their results discussed. By creating this section, a reader can easily refresh themselves as to what is each of these components, as that is crucial to understanding the results.

*Author's response*:

We understand the referee's point here and will edit the manuscript to include verbal descriptions of each of the annual cycles before they are defined in the model.

*Author's changes in manuscript*:

We will add the following paragraph to the top of Section 3:

"In this section we give five ways to define an annual cycle in the sea ice extent. We start with the traditional definition of the annual cycle and progressive define annual cycles that are more sophisticated and can represent more of the variation in the SIE over time. The second is an invariant annual cycle that retains the

365 day period of the traditional but incorporates the smooth functional form we might expect. The third adds amplitude variation to the invariant annual cycle so that the cycle itself varies from year to year with the amplitude of the year. The fourth adds phase variation to the invariant annual cycle, allowing it to capture the timing of the rise and retreat over each year. Finally, the fifth adds both amplitude and phase variation to the invariant annual cycle allowing it to represent variation over time in both the amplitude and phase of the SIE."

The section would also benefit from the results for each cycle being a new paragraph to ensure that they stand out, currently in most of the sections it runs straight from the methodology behind the cycle into its result. This runs the risk of the result being missed by readers.

*Authors' response:*

Agreed.

*Authors' changes in the manuscript:*

We have modified the text so that each cycle is distinct.

**Main Comments – Figures**

**Figure 1**

The smooth annual cycle as a blue line   is not distinguishable as blue. I would change the colour so that it can be resolved by the reader, even if that was black, which is what the line mostly appears to me currently (both on screen and in print). Figure 1 also sets a convention that Day 0 is the start of the cycle and Day 365 is the end of the cycle. However, the annual cycle is rarely exactly 365 days, it is "on average" but year on year it is not. How does this impact on both the figures and also the paper analysis? This should be addressed in the body text of the paper.

*Authors' response:*

In Fig. 1 the record for each year starts on Julian day 50 (the median minimum day). This is to address the length-of-cycle issue you raise (i.e., there are no missing or duplicated days in the plot). This choice is for ease of interpretation of Fig. 1. We will clarify in the text.

Outside the figure, the adjustment for the annual cycle differing in

length from year-to-year is precisely why the phase adjusted and amplitude and phase adjusted annual cycles are developed.

This also relates to a comment by Reviewer #1.

*Authors' changes in the manuscript:*

We will change the color of the smooth annual cycle to red and adjust the text around this figure to explain better what it represents.

**Figure 2**

Please label the panels as A and B. In general both panels are too small to properly resolve the detail within the image, particularly from the lines that overlap. In A, the green and black are nearly indistinguishable to my eye.

*Authors' response:*

Agreed

*Authors' changes in the manuscript:*

We will add (a)/(b) to the upper LHS corners and stack the panels (rather than side-by-side). We will increase the line width to make them easier to see.

**Figure 3**

The title over the figure appears to be incomplete. Given a title doesn't appear on the other figures is this an error for it to still remain?

*Authors' response:*

Agreed

*Authors' changes in the manuscript:*

We will remove the title.

**Figure 5 and Figure 6**

I'm not a fan of green, blue and black lines on the same figure, they are very hard to resolve, particularly the green and black which heavily overlap. The same issue also applied to Figure 6.

*Authors' response:*

Agreed

*Authors' changes in the manuscript:*

We will adjust the colors and line widths on Figures 5 and 6.

**Figure 7**

In a similar vein to Figures 5 & 6, the use of maroon and red on the same figure makes the figure harder to interpret. The figure legend also needs to be significantly bigger to make it more readable.

*Authors' response:*

Agreed

*Authors' changes in the manuscript:*

We will adjust the colors and legend size on Figures 7.

Specific (Line-by-Line) Comments

Line 16-17: The ordering of the references is ad-hoc, neither in publication date order or alphabetical order based on first author initial. In Line 20, they are ordered by date of publication, please adjust Line 16 to match.

*Authors' response:*

Agreed

*Authors' changes in the manuscript:*

We will re-order the references by date-of-publication throughout the paper.

Line 23: In the abstract (Line 2) you state that the SIE minima is in March; here in Line 23 you state it is late February. Please ensure consistency between these two dates. (See also Line 230-231 comments and Figure 1 comments).

*Authors' response:*

We will adjust the description of the typical SIE minima to be

February.

*Authors' changes in the manuscript:*

We will adjust the description of the typical SIE minima to be February.

**Line 24:** I would suggest altering this line to read: "The growth from minimum (trough) to maximum (peak) is slower than the retreat from maximum to minimum". Your use of trough refers to the minima, not the slope getting there, so I think defining both terms straight away fits better.

*Authors' response:*

We will alter the text as suggested, in the manuscript.

*Authors' changes in the manuscript:*

We changed the line to read "The growth from minimum (trough) to maximum (peak) is slower than the retreat from maximum to minimum".

**Line 39 (and others):** Some sections of the paper, such as here use an example refer-ence (Stine et al., 2009) is followed by "e.g.,". Elsewhere in the paper it precedes the reference (i.e. Line 29 "(e.g. Comiso and Steffen, 2001)"). I would prefer the Line 29 example, but either way is acceptable as long as it is consistent throughout the paper.

*Authors' response:*

The suffix version was due to a typo.

*Authors' changes in the manuscript:*

We changed paper to use the prefix version throughout.

**Lines 58-59:** The outline of the sections is not consistent with the body text. Results are in Section 3 alongside the Methods and Conclusions are in Section 4.

*Authors' response:*

We will correct the outline of the sections to make it consistent with the body of the text.

*Authors' changes to the manuscript:*

We corrected the outline of the sections to read: The data are described in Sect. 2, the model is defined and developed in Sect. 3. The results are presented and discussed in Sect. 3 while the Conclusions are made in Sect. 4.

**Line 81:** T = 2019. I was unsure if this should be T = 2019.000; T

= 2019.999 or something in between. Could this be clarified.

*Authors' response:*

2019 = 2019.000

*Authors' changes in the manuscript:*

We will change it to 2019.000

**Line 89:** I would appreciate a short (1 or 2 lines) explaining why you chose not to consider this data further. To raise it as a possible way to increase your temporal range and then simply dismiss it feels incomplete to me.

*Authors' response:*

We will alter the text to explain.

*Authors' changes in the manuscript:*

We changed the line to read: "However this requires a large and sophisticated model-based reconstruction and we do not further consider such methods in this paper.".

**Line 230-231:** Here the minima is listed a mid-February, which is different to the use of March (abstract) and late-February (i.e. Line 23). The consistency of the definition of the minima throughout the paper would be useful (albeit tricky due to the variability in the occurrence of the minima; as mentioned in the comments on Figure 1, this could do with being addressed earlier in the paper)

*Authors' response:*

We will adjust the description of the typical SIE minima to be February.

*Authors' changes in the manuscript:*

We will adjust the description of the typical SIE minima to be February.

**Lines 236-239:** Ice budget analysis work (i.e. Holland & Kwok, 2012; Holland, 2014; Pope et al., 2017; Holmes et al., 2019) has indicated that the surface winds play a role (advection and

divergence terms within the budget) throughout the whole of West Antarctica and into the Weddell Sea in both the advance and the retreat of sea ice. I would mention this work here with respect to the drivers of the advance and retreat of sea ice in addition to the modelling study mentioned.

*Authors' response:*

Thank you for these. We will incorporate the information from the mentioned studies into the discussion in lines 236-239.

*Authors' changes in the manuscript:*

**Line 239-240:** the sudden use of months (when most of the time dimension to date in the paper has been in days) is unnecessary and makes for awkward flow. I would remove 7.5 months and 4.5 months from these lines, as it would clean up the sentence which has to many "-" in it, making for a poor flow.

*Authors' response:*

We will remove the references to months and keep the reference to days.

*Authors' changes in the manuscript:*

We will remove the references to months and keep the reference to days.

Line 349: The authors should be praised for being so diligent in making their code accessible and open to peer review.

*Authors' response:*

We see the publication of code to be an essential part of the paper. Others can not reproduce our results without it. Also, others can start with this sophisticated code as a foundation and hence make much faster progress rather than reinventing work.

*Authors' changes in the manuscript:*

---

## Author Comment (AC2) · 18 Mar 2020

Summary:
This paper analyzes the seasonal cycle and interannual variability of Antarctic sea ice extent (SIE) using various statistically approaches. Different annual cycles are defined based on amplitude and phase. Variation in phase is found to explain more of the SIE variability, but combining both amplitude and phase explains substantially more variation than traditional methods. The approach shows that the low SIE extremes in 2016 were due mainly to a shift in phase.

**General Comment:**
This paper makes an interesting contribution to Antarctic SIE analyses. Two lingering questions with Antarctic SIE is the small positive trend that has been seen over the long-term satellite timeseries and the whiplash in recent years going from record highs to record lows within just a couple years. The analysis presented in this paper is unique for SIE and the paper brings to light many relevant characteristics of the SIE timeseries that provide insights into both short-term and long-term variability. For example, the idea that phase plays a more important role in the seasonal cycle than amplitude is revealing and seemingly important in better understanding the character and variability of Antarctic SIE. I have a few minor comments on various aspects. I recommend acceptance after minor revision to address these.

**Specific Comments (by line number):**
**1:** "troughing", while technically correct, reads awkwardly to me. Why not just say "reaches its minimum"? This occurs in a few other places in the text.

*Author's response*:
We will modify the text to say "reaches its minimum"

*Author's changes in manuscript*:

**23:** "peaking in September*. . .*and troughing in late February*. . .*on average.*"* Though you say on average, which is accurate, that masks a lot of variability. The minimum does sometimes occur in October and ranges from early Sept (even late Aug one year) through early Oct. The maximum can occur in March and ranges over 3 weeks. It might be worth providing a range along with the average to give a better sense of the variability, which as is shown later in the paper is important

*Author's response*:
To keep it simple in the Introduction we supply the median Julian days.

*Author's changes in manuscript*:
We added: "In Julian days, the median minimum day is 50 and the median maximum is 255."

**29:** Some of the day-to-day variation is also due to land-spillover (coastal effect of mixed land/water grid cells). It's not as variable as weather or changes in the ice cover, but I think it is important enough to warrant mention. (This is less of an issue in Antarctica because of the land ice along the coast, but still worth noting I think.)

*Author's response*:

We will include mention of this contribution to the variation.

*Author's changes in manuscript*: We will add:

"land-spillover (coastal effect of mixed land/water grid cells),"

**61-63:** The data reference is a little confusing. You say use the SMMR-SSMI-SSMIS Bootstrap Version 3 product, but reference Comiso (2017), which is the correct ref-erence. But you also reference Peng et al. (2013), and Meier et al. (2017), which refers to the NSIDC/NOAA Climate Data Record product. I understand the confusion here because the NSIDC/NOAA CDR does include the Bootstrap V3 concentrations

within the product. My assumption is that you used the Bootstrap V3 field within the NOAA/NSIDC CDR. So, I think all three references are warranted, but this could be more clearly explained, e.g., "We used the Bootstrap Version 3 concentration fields (Comiso, 2017) from the "NOAA/NSIDC Climate Data Record of Passive Microwave Sea Ice Concentration, Version 3" (Peng et al., 2013; Meier et al., 2017)." Or some- thing like that.

*Author's changes in manuscript*:

We will change the text to:

"We used the Bootstrap Version 3 concentration fields (Comiso, 2017) from the "NOAA/NSIDC Climate Data Record of Passive Microwave Sea Ice Concentration, Version 3" (Peng et al., 2013; Meier et al., 2017)."

**66:** You note that there are a number of days with no observations (in addition to the every-other-day SMMR). But one of those gaps is quite significant, with no data between early December 1987 and mid-January 1988. This is worth noting because it is unique in the record in terms of the length of the gap. Did you fill this in at all or leave the gap? Since the method doesn't require complete data, I assume not, but that should be made clear.

*Author's response*:

We will include mention of this contribution to the variation.

*Author's changes in manuscript*:

We will add: "In particular, there are no data between early December 1987 and mid-January 1988."

and later:

"As such we do not impute the missing days."

**69:** Day 0 is the minimum day of the year and then you just plot the next 364 days after that for each year. But of course, the date of the minimum differs from year to year. So, it seems like some years could have a gap – if the minimum of one year occurs before the minimum of the next year (i.e., >365 days between minimums) – where some data is not plotted, or conversely, you could have some data duplicated – if the minimum of one year occurs later than the minimum of the

next year (i.e., <365 days between minimums). Is this correct? Are these "missed" or "duplicated" accounted for in any way? Or does that potentially skew results at all?

*Author's response*:

In Fig. 1 the record for each year starts on Julian day 50 (the median minimum day). This is to address the length-of-cycle issue you raise (i.e., there are no missing or duplicated days in the plot). This choice is for ease of interpretation of Fig. 1. We will clarify in the text.

This also relates to a comment by Reviewer #2.

*Author's changes in manuscript*:

We will change the middle two sentences of this paragraph from "In this figure, day 0 on the horizontal axis represents the lowest SIE for the year, typically occurring around Julian day 50. We employ this convention for all of the time-series figures used in this paper."

to

"In this figure, day 0 on the horizontal axis represents the typical lowest SIE for the year, Julian day 50."

**Figure 3:** A few suggestions. First, the Day-to-day change is in Figure 4 (as noted in the Figure 3 caption). It seems like it is discussed in the context of Figure 4. So, is it necessary to include that line in Figure 3? Simpler is always better in my view, so one less line is helpful. And that would allow the y-axis to cover a smaller interval, which would more clearly show the variability lines. One thing that would be useful would be to label the max and min days (e.g., text with an arrow pointing to each). The day-to- day change does provide this, but it may not be immediately clear that the max occurs when the change turns from positive to negative. So, I think labeling would be helpful even if the day-to-day line is kept (but if labeling is included, then that line isn't really needed). The fonts on these figures are quite small – in the final version, they should be much better. Also, while the units are noted in the caption, it my view it is always better to include them with the axis labels. Similarly, for Figure 4.

*Author's response*:

We have chosen to retain the day-to-day line as it provides a detailed comparison with the variability. We have improved the figure in the other ways suggested.

*Author's changes in manuscript*:

We will add the arrows, increase the fonts size and add the units to the vertical axis.

**Figure 4:** There is an interesting feature in the traditional (orange) plot right around day 200, where the curve is less dense. All the other places have thin lines, highly varying day to day. But around day 200, there seems to be a period where the line just peaks and then declines over several days. Is that related to anything? Or is that just a quirk in the data, or just an optical illusion?

*Author's response*:

The region where the curve is less dense has two reasons. One is due to a quirk in the data. However, part of it is real and related to the relative stability of the ice extent in the region of the SIE maximum.

*Author's changes in manuscript*:

We will add a note in the text.

**206-214:** Why is the volatility higher for SMMR than for DMSP? Is it simply the every- other-day sampling? But there could also be an effect due to the sensor resolution (sensor footprint), which is actually smaller (higher resolution) in SMMR. I'm curious if the volatility of DMSP would match SMMR if every-other-day values were used from DMSP? Another, smaller aspect, is whether volatility changes from SSMI to SSMIS? If it's simply the temporal sampling, then I would expect there wouldn't be a change. But if there is a resolution component, then there might be a small effect since the sensor footprints are slightly different. While I think looking at that could be interesting, I guess it's not the main focus of the paper, so I can see not doing that. However, I think it is worth at least noting that the differences in volatility are due to temporal sampling (and maybe some resolution effect?), just to make that clear.

*Author's response*:

The model for the volatility is adjusted for the every-other-day sampling. We do not know the reason for the minor differences, but now add your speculation that they are due to the sensor resolution.

*Author's changes in manuscript*:

We will add a note in the text speculating on the sensor resolution change.

**244:** There is also more volatility at/near the maximum because there is more ice edge to vary. At the maximum, the perimeter of the ice cover is also at or near its maximum, which allows more areas to be affected by winds, currents, storms, etc.

*Authors' response:*
Thank you for this suggestion. We will include it in our discussion on page 244.

*Authors' changes in manuscript:*
We include in point 3, the suggestion that part of the increase in volatility at maximum may be due to the fact that the ice edge is larger.

**Figure 5:** What are the anomalies relative to – i.e., what is the base period? Likewise, for earlier figures, the y-axis should be labels with units.

*Author's response*:

The anomalies are relative to the annual cycles (in the legend) as defined in equation (14).

*Author's changes in manuscript*:

We will add a label to the vertical axis.

**281-291:** I understand the rationale for using the daily values over monthly values, but the advantage of monthly values is that you capture roughly the same period in the cycle – so you can look at trends near the maximum or near the minimum, which can be quite different than over a full year. But I also wonder is something is lost? – you're taking something with a big annual cycle and then just fitting trend lines through the entire 40 years. Would it make sense to do a Figure 6 for the max and min? Perhaps using the amplitude and phase

adjusted? Also, how does the curvilinear trend handle the endpoints – i.e., how does it calculate a trend from the beginning? In other words, how does the function (Eq. 15) calculate a smooth trend at the beginning of the time series? I assume that there is an endpoint fitting/smoothing, which may be in the equation. But some plain English explanation would be helpful as well.

*Author's response*:

It would make sense to do a Fig. 6 for the max and min. Indeed, it is natural to fit a non-parametric *quantile* regression curve for each quantile of the annual SIE distribution. The max and min curves are the extremes of this distribution. However, the analysis of these sets of curves would add substantial length and we will leave it for a subsequent paper.

A strength of Eq. 15 is that it directly incorporates the beginning and end of the time series into the smoothness equation.

*Author's changes in manuscript*:

We will add a note that in Eq. 15 "The last term also captures the beginning and end times smoothing."

**Figure 6:** What are the thin pink dashed lines? Are these just the beginning and end dates of the two periods? And the dashed line around the curvilinear trend?

*Author's response*:

The thin pink dashed lines demarcated the data segments (as Version 3 is cumulative). They were there for debugging purposes and will be removed.

The dashed lines are the 95% pointwise confidence bands for the smooth curvilinear trend equation.

*Author's changes in manuscript*:

We will remove the thin pink dashed lines and add a note on the confidence band.

288: The trend standard deviation (+/-) values should be included with the linear trend and maybe the trend significance.

*Author's response*:

We did not include the +/- value for the linear trend as they are not valid. They require the trend to be linear and the data indicate that it is curvilinear. Similarly, the trend is nominally significant.

*Author's changes in manuscript*:

We will add the text: "Were the trends linear they would be statistically significantly positive."

---

## Author Response (AR1)

The Cryosphere Discuss., https://doi.org/10.5194/tc-2019-203, 2019 © Author(s) 2019. This work is distributed under the Creative Commons Attribution 4.0 License.

**Point-by-point reply to the comments on "Modeling the annual cycle of daily Antarctic sea ice extent" by Mark S. Handcock and Marilyn N. Raphael**

Ted Maksym (Handling Editor) tmaksym@whoi.edu

Dear Ted,

We have submitted a final manuscript that includes each of the changes as we described in our responses to the two reviewers.

Attached below is a marked-up manuscript version showing the changes made in the LaTex files.

We have checked the manuscript for typos, etc.

We thank you for your work as editor for this manuscript.

Sincerely,

m. Handrock

Mark S. Handcock Professor of Statistics UCLA

Marily Roplace

Marilyn N. Raphael Professor of Geography UCLA

[revised manuscript text omitted]
(2)

where  $\sum_{t:doy(t)=s} 1 = 40$  is the number of years of data.

105 This traditional estimate,  $a_T[s]$ ,) has a number of statistical issues which reduce its utility for examining the sea ice variability. Firstly, it is typically based on data for a subset of the satellite era (e.g., from 1979 forward). Currently, this is about forty years of data, inducing intrinsic statistical variability into  $a_T[s]$  as an estimate of a[s]. This could be reduced by increasing the temporal range backward, by, for example, including data from the earlier satellite record (NIMBUS-5). Another option is to include information from proxy sources. However this requires a large and sophisticated model-based reconstruction and we

- 110 do not further consider such methods in this paper. We do not further consider these in this paper. Secondly,  $a_T[s]$  is computed separately for each day, ignoring the surrounding days. There is information in the temporally close days in the intuitive sense that days close to s, e.g., s - 1 and s + 1 will have similar values, albeit not exactly the same. This information is ignored by  $a_T[s]$ . Thirdly, we expect a[s]) to be smooth as a function of s so that changes in  $a_T[s]$  with s will be similar for days that are close. Fourthly, we expect that  $a_T[s]$  will "over fit" to the record making the estimated anomalies from it smaller than the
- 115 true anomaly,  $\alpha(t)$ , and the annual cycle estimates will be more variable than the true annual cycle. This last issue is induced by the finite record and the estimates of the anomaly  $\hat{\alpha}(t) = \operatorname{extent}(t) - a_T[\operatorname{doy}(t)]$  will be statistically different than those of  $\alpha(t)$ . In sum, the traditional estimate,  $a_T[s]$ , uses limited information, ignores other days, is not as smooth as we expect, due to day-to-day variation and it over fits to the record.

**Invariant Annual Cycle**

130

120 It is possible that smoothing the data could be a solution to the statistical issues that arise from the way in which the traditional annual cycle is calculated. To address this we define an *invariant annual cycle*,  $a_I[s]$ , which models a[s] as a cyclic cubic spline function (Wegman and Wright, 1983) of *s*. Specifically, a[s] is modeled as a piece-wise cubic polynomial that has a continuous second derivative, is continuous, has continuous 1st and 2nd derivatives at *T* and best fits the recorded (satellite-observed) extents while being smooth. The specific criterion for the last feature is to choose  $a_I[s]$  to minimize the penalized square error 125 (PSE):

$$PSE_{\lambda}(a) = \sum_{t=T_0}^{T} \{extent(t) - a[doy(t)]\}^2 + \lambda \int_{0}^{365} a''[s]^2 ds \qquad \lambda > 0$$
(3)

where a''[s] is the 2nd derivative of a[s] and  $\lambda$  is a smoothing parameter, chosen to balance the closeness of fit to the recorded values (the first term) with the smoothness of a[s] (the second term). Hence, choosing the function a[s] that minimizes  $PSE_{\lambda}(a)$ provides a balanced representation of the annual cycle. It prioritizes smoothness of a[s] over the closeness of fit of a[s] to the recorded extents. Note that the traditional estimator,  $a_T[s]$ , is the minimizer with  $\lambda = 0$ , that is, with no penalty for lack of smoothness. The choice of  $\lambda$  is subjective. In this work we choose to maximize the ability to predict unrecorded extents. Specifically, we use Generalized Cross Validation (GCV) (Craven and Wahba, 1978) to choose, and the R package mgcv by Simon Wood for analysis (Wood, 2004, 2017). The annual cycle so obtained is the optimal smoothest annual cycle chosen to minimize the mean squared error (MSE) of SIE. Any trends are removed and there is no adjustment for phase or amplitude.

135 Fig. 2(a) compares the traditional annual cycle (plotted from Julian day 50 in 2016 to day Julian day 49 in 2017), with the recorded SIE, and the invariant annual cycle. The visual improvement is modest but, as shown in Table 1, the invariant annual cycle represents a 29.8% improvement in the MSE compared to the traditional. Note that both annual cycles overestimate the SIE in the retreat phase of the ice.

**Amplitude Adjusted Annual Cycle**

- 140 The invariant annual cycle has the same motivation as the traditional annual cycle while being a clear statistical and conceptual improvement over the traditional. However, we argue that since it is also fixed by day-of-year, it may be too restrictive since it, like the traditional, disguises the contributions of both amplitude and phase to the annual cycle. To address this we define a complementary annual cycle that is deformed each year in two ways. The first is *amplitude* in the sense that the yearly maximum and minimum extents may vary but the *shape* of the daily extent may be invariant. We enable the annual cycle to
- 145 vary from year-to-year as a parametrized function of the annual cycle shape function. Specifically, we define the *amplitude adjusted annual cycle*,  $a_A[s, y]$  to satisfy:

$$extent(t) = a_A[doy(t), min \cdot extent(year(t)), max \cdot extent(year(t))] + \alpha(t)$$
(4)

where

$$a_A[s,\min,\max] = u_A[s](\max-\min) + \min$$
(5)

and year(t) is the year for t (e.g., 2010), max-extent(y) is the scale parameter giving the maximum extent for year y and  $\min$ -extent(y)) is the scale parameter giving the minimum extent for year y. Here  $u_A[s]$  is an invariant annual cycle for the standardized extent. It is defined in an analogous way to the invariant annual cycle as a smooth function. Specifically,  $u_A[s]$  as a cyclic cubic spline function of s chosen to minimize the penalized square error:

$$PSE_{\lambda_A}(u) = \sum_{t=T_0}^{T} \left\{ \frac{\operatorname{extent}(t) - \min\operatorname{extent}(\operatorname{year}(t))}{\max\operatorname{extent}(\operatorname{year}(t)) - \min\operatorname{extent}(\operatorname{year}(t))} - u[s] \right\}^2 + \lambda_A \int_0^{365} u''[s]^2 ds \qquad \lambda_A > 0 \tag{6}$$

155 where  $\lambda_A$  is a smoothing parameter with the same role as  $\lambda_I$ .

This annual cycle gives a different decomposition of the extent than the invariant annual cycle as it captures variation due to amplitude variation. Specifically, adjusting for amplitude results in a 55.2% improvement in the MSE compared to the traditional (See Table 1). Note that this allocates that component of the variation in extent due to amplitude variation to the annual cycle rather than the residual term,  $\alpha(t)$  (See Eq. (4)). The magnitude of the change clearly underscores the importance of amplitude variations in the definition of the annual cycle.

**Phase Adjusted Annual Cycle**

160

165

Another component of the annual cycle that is important is the phase. This is the timing of the maximum and minimum extents. It is important because it determines the length of the annual cycle and influences its shape. We enable the annual cycle to vary from year-to-year as a parametrized function of the phase of the annual cycle shape function, defining the *phase adjusted annual cycle*,  $a_P[s]$ :

$$extent(t) = a_P[phase(t)] + \alpha(t)$$
(7)

---

## Author Response (AR2)

> Editor Decision: Publish subject to technical corrections (06 May 2020) by Ted Maksym

> Comments to the Author:

> Just a few very minor I found, that at most require a few words to clarify:
>
> Figure 1 fonts are small.

Increased by a factor of 2.
>
> Line 29 — remove comma; remove excess parentheses.

done

>
> Line 33 — add "and" before land

done
>
> Line 57 — comma before rather

done

>
> Line 128 and figure 2. I found this section a bit confusing and had to read a few times. You are comparing the traditional and invariant annual cycles with the actual recorded observations for 2016-2017. Since the traditional and invariant are both versions of the typical annual cycle for the 40-year period, is it not unsurprising that they should not compare well with 2016-2017? The improvement in MSE is surprising, since it is hard to tell the traditional and annual cycles apart from these plots, but I don't understand why we would expect a particular year to match the "mean" annual cycle all that well, or why we might expect the invariant to fit better. Is it not possible that for some years, the invariant might actually fit worse? It is clear that the APAC matches closely, but the reduction in MSE is less than I might have expected compared to the invariant based on Fig 2. Perhaps it is just that the first half of the year matches quite well for all, so the actual MSE is relatively small in all cases. Maybe just some short statement to this effect would help?

We do not expect a particular year to fit well the traditional or invariant. Indeed, 2016 is not even a typical year. We included 2016 to show the scale of deviation of the invariant from the traditional compared to the scale of year-to-year variation. Visually, the invariant does not fit much better and for some years it will be worse. However, taken over the for satellite recorded era the invariant is 28.7% closer to the individual years compared to the traditional. The APAC is a significant improvement over the traditional and invariant in terms of RMSE (0.576, 0.481, 0.272, respectively). In this metric the, APAC is over three times the improvement of the invariant over the traditional.

>
> Line 201-204 — isn't the difference in true vs recorded sea ice extent just due to errors in the retrieval (algorithm, weather, etc), and not real variations such as due to sea ice drift? Perhaps you mean that short term variability between the different passes, or timing of passes can create errors due to ice movement (i.e. when the daily extents are averaged from two daily swaths). I suppose real physical variations in ice conditions due to drift or other factors at the ice edge could lead to small changes in the algorithm's

calculated concentration; when this variation occurs when C ~ 15%, then you could get pixels switching between ice and no ice when binarized for ice extent that would contribute to volatility. My only point here is to state why ice drift effects might lead to variations that are not real. Similarly in the conclusion, you conflate the volatility with ephemeral effects, yet there are ephemeral effects that are real day-to-day changes in sea ice extent. I suggest a parenthetical definition of what you mean by ephemeral when this is mentioned in the conclusion.

You make an important point and a subtle one. Statistical, we unambiguously specify the volatility in equation (13). But what are the sources of this variation? Clear sources are the errors in retrieval (algorithm, weather, etc). The definition of SIE, however arbitrary, includes the sea ice drift, so it is a non-source. The more subtle sources are the interaction between the errors in retrieval and weather (that is, how weather effects the quality of the retrieval). Weather can influence both the SIE and the volatility through the interaction between weather and the satellite effects and algorithm artifacts. Another source is timing. We represent the SIE daily, but it temporally continuous so that the timing of the satellite passes, however individually accurate, leads to variation. So these are sources for the volatility also. We have edited the conclusion to help a bit here and have replaced the whimsical "ephemeral" with "transient".

>
> Figure 6 — fonts are very small (also, please check font sizes in other figures so they will all be readable when figures are sized for publication).

We edited them all.

>
> Line 382 — "writing and editing"

done

In addition, we made minor edits to the "Code and data availability" section (line 375) to indicate the code is (rather than will be) available. The cite will go live on publication.